# Impact of Multidisciplinary Team Management on Survival and Recurrence in Stage I–III Colorectal Cancer: A Population-Based Study in Northern Italy

**DOI:** 10.3390/biology13110928

**Published:** 2024-11-15

**Authors:** Lucia Mangone, Maurizio Zizzo, Melissa Nardecchia, Francesco Marinelli, Isabella Bisceglia, Maria Barbara Braghiroli, Maria Chiara Banzi, Angela Damato, Loredana Cerullo, Carlotta Pellegri, Fortunato Morabito, Antonino Neri, Massimiliano Fabozzi, Carmine Pinto, Paolo Giorgi Rossi

**Affiliations:** 1Epidemiology Unit, Azienda USL-IRCCS di Reggio Emilia, 42123 Reggio Emilia, Italy; francesco.marinelli@ausl.re.it (F.M.); isabella.bisceglia@ausl.re.it (I.B.); mariabarbara.braghiroli@ausl.re.it (M.B.B.); paolo.giorgirossi@ausl.re.it (P.G.R.); 2Unit of Surgical Oncology, Azienda USL-IRCCS di Reggio Emilia, 42123 Reggio Emilia, Italy; maurizio.zizzo@ausl.re.it (M.Z.); melissanardecchia@hotmail.com (M.N.); massimiliano.fabozzi@ausl.re.it (M.F.); 3Medical Oncology Unit, Azienda USL-IRCCS di Reggio Emilia, 42123 Reggio Emilia, Italy; maria.banzi@ausl.re.it (M.C.B.); angela.damato@ausl.re.it (A.D.); carmine.pinto@ausl.re.it (C.P.); 4Quality Office, Azienda USL-IRCCS di Reggio Emilia, 42123 Reggio Emilia, Italy; loredana.cerullo@ausl.re.it (L.C.); carlotta.pellegri@ausl.re.it (C.P.); 5Gruppo Amici Dell’Ematologia Foundation—GrADE, 42123 Reggio Emilia, Italy; fortunato.morabito@grade.it; 6Scientific Directorate, Azienda USL-IRCCS di Reggio Emilia, 42123 Reggio Emilia, Italy; antonino.neri@ausl.re.it

**Keywords:** colorectal cancer, stage, multidisciplinary team, recurrence, disease-free survival, overall survival, population-based study

## Abstract

Colorectal cancer is a frequent neoplasm in the general population in both males and females. If oncological screening allows for the early diagnosis of the neoplasm, when present, the management of tumors that are not detected at an early stage becomes more complex. In recent years, in the most cutting-edge hospitals, the establishment of dedicated therapeutic paths that include a team of professionals from different specialties (oncologist, surgeon, radiotherapist, pathologist, radiologist, and, frequently, psychologist and data manager) has become increasingly common. This work shows how, in a province of Northern Italy, the presence of a multidisciplinary team dedicated to the management of patients with colorectal cancer is able to best deal with the patient’s recurrences and take care of the person throughout the disease’s complexity. In the era of personalized medicine, personalization involves not only drugs but also the management of the person and not just the tumor.

## 1. Introduction

Colorectal cancer (CRC) ranks as the third most common malignancy globally, with over 1.9 million new cases diagnosed annually, contributing to more than 900,000 cancer-related deaths in 2022 [1]. The pathogenesis of CRC often follows the adenoma–carcinoma sequence, and early detection through oncological screening has had a profound impact on both the disease incidence and mortality [2,3]. Early-stage diagnosis has been associated with a significant increase in survival rates [4,5,6], while advancements in pharmacological interventions for metastatic disease have contributed to a decrease in mortality, even for patients diagnosed at advanced stages [7,8].

Currently, the 5-year survival rate for CRC stands at approximately 65% for both males and females, which is a figure that has progressively improved due to enhanced screening efforts [9,10,11], therapeutic innovations [12,13,14], and the development of multidisciplinary team (MDT) approaches in patient management [15,16,17]. The MDT approach has emerged as a crucial factor in reducing recurrence rates and mortality, as well as improving patient outcomes by fostering a collaborative, evidence-based decision-making process and ensuring the appropriateness of treatments. MDTs facilitated the collegial review of individual patient cases to establish diagnostic strategies, confirm diagnoses, and determine optimal treatment modalities [18]. This approach has been associated with improved surgical appropriateness [19] and the more accurate administration of adjuvant chemotherapy [20]. Evidence in the literature suggests that MDT presentation correlates with improved survival outcomes, especially with advanced-stage CRC [21,22,23,24]. However, despite the presence of MDTs in referral cancer centers, not all CRC cases are evaluated by MDTs. Factors such as advanced age at diagnosis, comorbidities, early mortality, and the presence of superficial tumors amenable to endoscopic treatment (requiring no further intervention) have been associated with a lower likelihood of MDT assessment and follow-up [18]. In recent years, the proportion of patients followed by MDTs has notably increased. For instance, in France, MDT involvement rose from 32% in 2000 to 82% in 2018 [25,26].

The objective of this study is to evaluate whether MDT-managed CRC patients in a province of northern Italy experience different outcomes compared to those not managed by MDTs, specifically in terms of disease-free survival (DFS) and overall survival (OS).

## 2. Materials and Methods

### 2.1. Data Sources

The Reggio Emilia (RE) Cancer Registry (CR) covers a population of 532,000 inhabitants and has been systematically registering incident cases since 1996. It maintains up-to-date records, with incidence data extending through the end of 2021. The registry boasts a high rate of microscopic confirmation, with 93.2% of CRC cases confirmed via histological examination, and a minimal percentage of death certificate-only (DCO) cases (<0.1%) [27]. The primary data sources for the registry include microscopic reports, hospital discharge records, and mortality data. These are supplemented with laboratory results, diagnostic imaging, and information provided by general practitioners. In this way, we try to ensure maximum completeness (not losing cases) and diagnostic accuracy (high percentage of microscopic confirmations and low percentage of DCO). The CR adheres to established collection protocols aimed at generating accurate statistics on cancer incidence, mortality, prevalence, and survival rates for the resident population, which is in line with the requirements of epidemiological reporting. This framework is governed by Italian Law No. 29 of 03/22/2019, which regulates the CRs and exempts them from obtaining informed consent for data collection. The protocols for epidemiological analyses conducted using RECR data have been approved by the provincial Ethics Committee of Reggio Emilia (protocol no. 2014/0019740 of 04/08/2014).

This study included all incident cases of stage I–III CRC diagnosed between 2017 and 2018. CRC cases were defined according to the International Classification of Diseases for Oncology, Third Edition (ICD-O-3) as topography codes C18–C19 [28]. Information regarding tumor stage based on the TNM classification system (8th edition) [29], and recurrence data were obtained through a review of medical records from the hospital. All tumor incident cases in 2017–2018 were included and then linked with the MDT data; in this way we only reported what was actually observed among the cases of colorectal cancer, dividing them into MDT and no-MDT.

### 2.2. Multidisciplinary Team

In 2017, a comprehensive review and update of diagnostic and therapeutic protocols for major cancer sites were implemented in Reggio Emilia. This initiative aimed to establish evidence-based decision-making processes and clearly define roles, responsibilities, and coordination among services involved in patient care. As part of this initiative, the main steps in the care pathways for cancer patients are systematically discussed by site-specific multidisciplinary teams (MDTs). The implementation of these new protocols began with breast cancer in 2005, and the MDT for CRC was established in 2015. The MDT includes medical oncologists, endoscopists, pathologists, radiologists, nuclear medicine specialists, and surgeons. In addition, each MDT is attended by an administrator and a data manager, with psychologists and nutritionists participating as needed. MDT meetings are held weekly to determine the most appropriate therapeutic strategy for each patient, taking into account factors such as age, social status, and tumor characteristics. The entire process is monitored through quality indicators that assess both process and early outcome measures, with feedback provided annually to all healthcare professionals involved in patient care.

### 2.3. Statistical Analysis

Descriptive statistics were calculated for all patients included in this study (n = 460) and for a subgroup of patients who were alive 90 days after diagnosis and had no recurrence within 6 months of diagnosis (n = 414) [30]. Patient characteristics, including age at diagnosis (categorized into three groups: <50, 50–69, and over 69 years), gender, tumor site (colon vs. rectum), stage (I, II, III), surgical intervention, and chemotherapy, were reported and stratified by MDT vs. non-MDT management groups. The differences between the groups were evaluated using Fisher’s exact and χ^2^ tests, as appropriate. A multivariable Cox proportional hazard regression model was constructed for the restricted subgroups to assess the associations among stage, age at diagnosis, gender, MDT involvement, recurrence, and overall survival and disease-free survival.

Time-to-event outcomes, such as DFS and OS, were calculated using the Kaplan–Meier method. DFS was defined as the time from tumor resection to either the diagnosis of recurrence or death from any cause. Hazard ratios (HRs) were reported with 95% confidence intervals (CIs). All statistical analyses were performed using STATA 16.1 software (StataCorp LLC, College Station, TX, USA).

## 3. Results

During the period of 2017–2018, a total of 460 patients with stage I–III CCR were included in this study. Of these, 300 patients (65.2%) were managed by MDT, while 160 patients (34.8%) did not receive MDT management (Table 1). The majority of patients were aged 70 years or older (62.2%), were male (57.2%), had colon cancer (73.3%), and were diagnosed with disease stage II (36.1%). Approximately two-thirds of cases underwent surgery (74.4%), and 27% received chemotherapy. Compared to the non-MDT group, patients in the MDT group were younger, had a higher proportion of rectal tumors, had fewer locally advanced tumors, and more frequently underwent both surgery and chemotherapy.

Table 2 presents data on patients in whom recurrence was analyzed, excluding those who experienced recurrence within 6 months of diagnosis or died within 3 months. Age and sex distributions were similar between the MDT and non-MDT groups. However, a higher proportion of stage I cancers was observed in the MDT group compared with the non-MDT group (38.3% vs. 23.5%, *p* < 0.05). In the MDT group, 24 recurrences (8.5%) were recorded, with 4 occurring in stage I and 20 occurring in stages II–III. The majority of metastases in the MDT group involved the lymph nodes and peritoneum and occurred between 6 and 12 months after diagnosis. In contrast, the non-MDT group recorded 13 recurrences, all in stage II–III, with most metastasis (nine) affecting the liver and lungs and occurring mainly after 12 months from diagnosis.

Table 3 shows that the DFS was worse for patients with stage III cancer (HR 2.75; 95% CI 1.72–4.41), patients aged 70 years and older (HR 4.08; 95% CI 1.29–12.91), and those in the non-MDT group (HR 1.62; 95% CI 1.12–2.32). These same variables also negatively affected OS, with particularly poor outcomes for patients aged 70 and older (HR 5.32; 95% CI 1.31–21.67) and those in the non-MDT group (HR 1.85; 95% CI 1.26–2.71). No statistically significant differences in DFS and OS were observed between patients diagnosed in 2017 and 2018.

The Kaplan–Meier curves demonstrated a significant difference in survival between the MDT and non-MDT groups. At 3 years, the DFS rate was 78% (95% CI 73–83%) in the MDT group compared to 65% (95% CI 56–73%) in the non-MDT group (Figure 1A). Similarly, survival was better in the MDT group, with a 3-year OS rate of 83% (95% CI 78–87%), compared to 69% in the non-MDT group (95% CI 60–76%) (Figure 1B).

## 4. Discussion

This study aimed to evaluate whether management by an MDT results in different outcomes compared to non-MDT management in terms of recurrence and survival in CRC patients in a province of northern Italy.

From 2017 to 2018, a total of 460 patients with stage I–III CRC were analyzed, of which 300 (65%) were managed by MDTs. This study revealed that patients followed by MDT were generally younger, had an earlier stage of disease, had a higher proportion of rectal cancers, and more often underwent surgery and chemotherapy compared to non-MDT patients. Additionally, outcomes were more favorable for patients followed by MDTs, both in terms of DFS and OS, which is consistent with findings from previous research.

Although our study did not explore the exact mechanisms linking MDT management to improved outcomes, previous studies have indicated that MDT involvement enhances treatment adherence, improves survival rates, and reduces hospital readmission rates in CRC patients [31]. Even by exploring only the rectal pathology, patients followed by MDTs have better outcomes both in terms of OS and whether it is a curative (neoadjuvant CT + surgery) or palliative approach [32]. However, the integration of MDTs into cancer management has not been universally implemented, with variations depending on geographic and cultural factors [33,34,35]. The multidisciplinary approach has been shown to positively impact outcomes in CRC, primarily through early diagnosis and comprehensive treatment strategies [36,37]. The success of MDT care is attributed not only to the presence of key medical specialists, such as oncologists, pathologists, surgeons, and radiologists, but also to the support of psychologists and technical and administrative personnel. Initially, MDTs were more likely to manage less severe cases that were diagnosed at earlier stages [38], which is a pattern that was similarly observed in a Spanish study [39]. In Jung’s study accounting for 1383 CRC patients, MDT involvement influenced final treatment decisions in up to 13% of cases and modified radiological outcomes in nearly half of cases [40].

In our study, we observed a recurrence rate of 8.5%, with no significant differences between the MDT and non-MDT groups. This rate is notably lower than the 18% recurrence rate reported by Martinez for stage II and III tumors, underscoring the association of T4, N2 status, and tumor budding with an unfavorable prognosis [41]. Recurrence rates are clearly influenced by the stage of the disease. Our findings indicated a recurrence rate of 3.7% for stage I tumors, aligning with the 4.4% reported by Nors [42], 2.9% reported by Paik [43], and 3–8% reported by Qaderi [44], in patients undergoing surgery. Additionally, Nikoolic reported a recurrence rate of 38% after 3 years in surgical cases, highlighting the presence of unfavorable prognostic factors, such as advanced age, mucinous histotype, high Ca 19.9 values, stage III disease, and male sex [45]

Notably, advanced tumor stages were more strongly associated with an increased risk of recurrence. Quaderni reported recurrence rates of 12%, 16%, and 24% for stage II cancer and 31%, 24%, and 38% for stage III cancer in the right colon, left colon, and rectum, respectively [44].

In our study, multivariable analysis revealed a worse DFS in stage II patients, individuals aged over 79 years, and those not followed by an MDT. OS mirrored these findings, with non-MDT patients demonstrating a roughly two-times increased risk of recurrence (HR = 1.85), suggesting that a management optimization through an MDT approach may, at least in part, prevent recurrences. At 3 years, the DFS was 78% in the MDT group compared to 65% in the no-MDT group. These values are favorable when compared with surgical patients, who exhibited a DFS of 81.4% for stage II and 49% for stage III [46]. OS at 3 years was 83% in the MDT group vs. 69% in the non-MDT group. These findings are consistent with the Rosander study, which reported 3-year OS rates of 80% in the MDT group and 68% in the non-MDT group [47]. Accordingly, a Swedish study also found that for patients with locally advanced CRC, survival was significantly higher (80%) in the MDT group compared to the non-MDT group (68%) at both 3 and 5 years (73% and 60%, respectively) [47]. The same results were confirmed at 5 years. Similarly, Munro confirmed these results at 5 years, with survival rates of 63.1% in patients who were enrolled in an MDT program, which is in contrast with an OS rate of 48.2% for those who did not, particularly emphasizing the benefits for more advanced tumor stages (adjusted HR for advanced disease: 0.65; CI: 0.45 to 0.96, *p*  =  0.031) [22].

In conclusion, our province, like other regions in northern Italy, is characterized by a high incidence of cancer across all sites [48] but also demonstrates strong outcomes linked to high adherence to oncological screening programs [49], contributing to reductions in both CRC mortality and incidence [3]. Access to care through the Comprehensive Cancer Center and MDT involvement has become increasingly widespread, improving patient outcomes.

The strengths of this study lie in its population-based approach and the use of a cancer registry, minimizing selection bias. Additionally, this study involved thorough data retrieval on tumor stage, treatment, and recurrence through direct consultation of the medical records. We are aware that the data from a single CR cannot be generalized to the entire country but can be representative of what happens in the regions of northern Italy, both in terms of tumor incidence and appropriateness of treatment.

This work confirms, using population data, what has already been observed at the hospital level, as follows: Although the MDT is not able to prevent recurrence, patients with recurrence followed by an MDT have better outcomes both in terms of DFS and, above all, overall survival, demonstrating the fact that the MDT is always able to ensure at least a better management. Furthermore, the small sample size due to this study’s single-center design limits its generalizability. Expanding this study to additional centers with robust CR and MDT programs could help validate the findings.

## 5. Conclusions

The implementation of MDTs in colorectal cancer care is a complex and resource-intensive process that requires not only specialized professionals but also strong technical and administrative support. While colorectal cancer benefits from well-established screening and early diagnosis protocols, a multidisciplinary approach is essential for patient management, including cases diagnosed outside of screening programs.

This study highlights the positive impact of MDT, showing an improvement in both DFS and OS. MDT likely ensures appropriate staging and treatment, leading to better management of recurrences. Although recurrences remain a challenge, MDT plays a crucial role in optimizing patient care, even in advanced cases.

## Figures and Tables

**Figure 1 biology-13-00928-f001:**
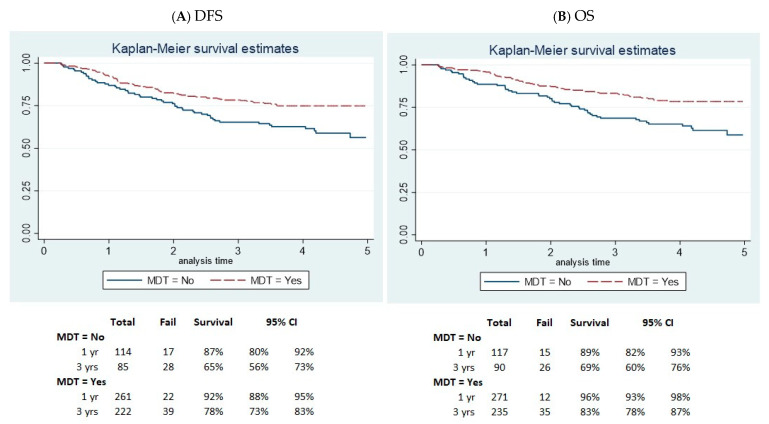
Kaplan–Maier curve of disease-free survival (**A**) and overall survival (**B**) by MDT and non-MDT group.

**Table 1 biology-13-00928-t001:** Colorectal cancer, stage I–III, years 2017–2018: characteristics of patients.

			MDT				*p*-Value
			Yes		No		
	n	%	n	%	n	%	
Overall	460	100	300	65.2	160	34.8	
Age at diagnosis (group)							
<50	25	5.4	19	6.3	6	3.8	
50–69	149	32.4	100	33.4	49	30.6	0.37
70+	286	62.2	181	60.3	105	65.6	
Year							
2017	244	53.0	136	45.3	108	67.5	<0.01
2018	216	47.0	164	54.7	52	32.5	
Gender							
Male	263	57.2	171	57.0	92	57.5	0.92
Female	197	42.8	129	43.0	68	42.5	
Site							
Colon	337	73.3	208	69.3	129	80.6	<0.01
Rectum	123	26.7	92	30.7	31	19.4	
Stage							
I	146	31.7	110	36.6	36	22.5	<0.01
II	166	36.1	101	33.7	65	40.6	
III	148	32.2	89	29.7	59	36.9	
Surgery							
Yes	342	74.4	235	78.3	107	66.9	<0.05
No	85	18.5	44	14.7	41	25.6	
Unknown	33	7.1	21	7.0	12	7.5	
Chemotherapy							
Yes	124	27.0	89	29.7	35	21.9	0.20
No	303	65.9	190	63.3	113	70.6	
Unknown	33	7.1	21	7.0	12	7.5	

**Table 2 biology-13-00928-t002:** Colorectal cancer, stage I–III, years 2017–2018: characteristics of patients and recurrences by stage, site, and months (excluding patients who experienced recurrence within 6 months of diagnosis or died within 3 months).

	MDT				
	Yes (n = 282)		No (n = 132)		*p*-Value
	n	%	n	%	
Age at diagnosis (group)					
<50	18	6.4	6	4.5	0.71
50–69	99	35.1	45	34.1	
70+	165	58.5	81	61.4	
Gender					
Male	162	57.5	72	54.5	0.58
Female	120	42.5	60	45.5	
Stage					
I	108	38.3	31	23.5	<0.05
II	96	34.0	59	44.7	
III	78	27.7	42	31.8	
Recurrences					
Yes	24	8.5	13	9.9	0.56
No	255	90.4	116	87.9	
Unknown	3	1.1	3	2.3	
Only patients with recurrence	Yes (n = 24)		No (n = 13)		*p*-value
	n	%	n	%	
Stage					
I	4	3.7	0	0.0	0.28
II–III	20	11.5	13	12.9	
Site					
Liver	8	33.3	6	46.2	0.73
Lung	5	20.8	3	23.1	
Lymph nodes	3	12.5	0	0.0	
Peritoneum	8	33.3	4	30.7	
Months					
6–12	11	45.8	4	30.8	0.38
12+	13	54.2	9	69.2	

**Table 3 biology-13-00928-t003:** Colorectal cancer, stage I–III, years 2017–2018: Cox regression analysis, adjusted for age, stage, gender, year, and MDT for disease-free and overall survival.

Characteristics	Disease-Free Survival			Overall Survival		
	HR	95% CI	*p*-Value	HR	95% CI	*p*-Value
Stage						
I	1.00	Ref.		1.00	Ref.	
II	1.30	0.79–2.14	0.30	1.16	0.69–1.97	0.57
III	2.79	1.74–4.49	<0.01	2.50	1.52–4.10	<0.01
Age at diagnosis						
<50	1.00	Ref.		1.00	Ref.	
50–69	1.36	0.40–4.57	0.62	1.52	0.35–6.61	0.58
70+	4.09	1.29–12.98	<0.05	5.23	1.28–21.35	<0.05
Gender						
Female	1.00	Ref.		1.00	Ref.	
Male	1.14	0.80–1.64	0.47	1.06	0.72–1.56	0.77
MDT						
Yes	1.00	Ref.		1.00	Ref.	
No	1.66	1.14–2.41	<0.01	1.96	1.31–2.91	<0.01
Year						
2017	1.00	Ref.		1.00	Ref.	
2018	1.19	0.82–1.75	0.36	1.32	0.88–1.98	0.19

## Data Availability

The data presented in this study are available on request from the corresponding author. The data are not publicly available due to ethical and privacy issues; requests for data must be approved by the Ethics Committee after the presentation of a study protocol.

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
