# Peer review of "Impact of Multidisciplinary Team Management on Survival and Recurrence in Stage I–III Colorectal Cancer: A Population-Based Study in Northern Italy"

_biology, 2024, doi:10.3390/biology13110928_

Round 1
Reviewer 1 Report
Comments and Suggestions for Authors
The manuscript by Mangone et al showed multidisciplinary team management improves outcomes in patients with stage I-III colorectal cancer compared to non-MDT care in a northern Italian province. The research design is rational and result is reasonable.
1. Table 1-3 need to be improved, it is hard to read.
2. Figure 1 need to be redrawn, the font is too small, and the color and lines are not clear.
Author Response
The manuscript by Mangone et al showed multidisciplinary team management improves outcomes in patients with stage I-III colorectal cancer compared to non-MDT care in a northern Italian province. The research design is rational and result is reasonable.
RE: Thank you so much for the positive comment.
- Table 1-3 need to be improved, it is hard to read.
RE: Thanks for the advice, we have changed the layout of table 1 and added the p-values to table 1 and 3.
- Figure 1 need to be redrawn, the font is too small, and the color and lines are not clear.
RE: Thanks for the comment, we have replaced the figure, hoping that it is now easier to understand.
Submission Date
18 September 2024
Date of this review
07 Oct 2024 16:37:09
Reviewer 2 Report
Comments and Suggestions for Authors
The authors conducted a population-based cohort study to investigate the prognostic impact of multidisciplinary team (MDT) management on oncological outcomes in patients with stage I-III colorectal cancer from 2017 to 2018.
While the topic is compelling, significant criticisms need to be addressed.
Comments:
1. The primary concern of the study is selection bias. The authors should clarify the substantial differences in baseline patient and tumor characteristics between the MDT and non-MDT groups.
2. Tables 1 and 2 should be combined, with a P value included for each variable.
3. A column for P values should be added to Table 3.
4. This study demonstrates a strong association between MDT and key prognostic factors, including disease stage. To establish the independent prognostic effect of MDT, the authors should conduct a multivariate Cox regression analysis and present the findings in a separate table.
Author Response
The authors conducted a population-based cohort study to investigate the prognostic impact of multidisciplinary team (MDT) management on oncological outcomes in patients with stage I-III colorectal cancer from 2017 to 2018.
While the topic is compelling, significant criticisms need to be addressed.
RE: Thanks for the comments, we hope the revisions made are compliant.
Comments:
- The primary concern of the study is selection bias. The authors should clarify the substantial differences in baseline patient and tumor characteristics between the MDT and non-MDT groups.
RE: Thank you for the request because it allows us to better clarify how we selected the population. The study is population-based because it collects all cases of malignant colorectal tumors occurring in the province of Reggio Emilia in 2017-2018. For this reason, there is no selection bias because we did not select MDT and non-MDT patients a priori. What we did instead was the reverse path, that is, observing the number of tumors occurring in 2017 and describing, in a completely random way, those that were followed by MDT and not. The same work was done for 2018. Thank you for the request because it allowed us to better explain the concept in the material and methods section.
- Tables 1 and 2 should be combined, with a P value included for each variable.
RE: We added the missing p-values as requested by the reviewer. However, we do not feel comfortable combining the two tables because the population studied is different: the first table includes 460 patients, i.e. all cases of colorectal cancer. The second table, instead, includes 414 cases because patients with recurrence within 6 months of diagnosis or death within 3 months were excluded. Furthermore, the second table contains data on recurrence, which is one of the outcomes of our study.
- A column for P values should be added to Table 3.
RE: Thanks, we have added the p values as requested.
- This study demonstrates a strong association between MDT and key prognostic factors, including disease stage. To establish the independent prognostic effect of MDT, the authors should conduct a multivariate Cox regression analysis and present the findings in a separate table.
RE: Thanks for the request, Table 3 shows two Cox regression models for both overall survival and disease free survival and the MDT variable is included in both models. We have changed the table caption to clarify.
Reviewer 3 Report
Comments and Suggestions for Authors
The study on "Impact of Multidisciplinary Team Management on Survival 2 and Recurrence in Stage I-III Colorectal Cancer: A Population- 3 Based Study in Northern Italy" performed study on 460 patients. To me, this is not an innovative study which contributes significantly towards the early diagnosis and prompt therapeutic approach of CRC.
Author Response
The study on "Impact of Multidisciplinary Team Management on Survival 2 and Recurrence in Stage I-III Colorectal Cancer: A Population- 3 Based Study in Northern Italy" performed study on 460 patients. To me, this is not an innovative study which contributes significantly towards the early diagnosis and prompt therapeutic approach of CRC.
RE: Thank you for your comment. We are aware that the study does not report significantly innovative data compared to what is reported in the literature. However, the possibility of working with data that come from a population-based tumor registry, without selection bias, could represent a stimulus on the real usefulness of taking charge of the patient by an MDT. In fact, although the MDT is not able to prevent recurrence, patients with recurrence followed by an MDT have better outcomes both in terms of DFS, but above all of Overall Survival, demonstrating the fact that the MDT is always able to ensure at least a better management. We have better specified this concept in the conclusion.
Round 2
Reviewer 2 Report
Comments and Suggestions for Authors
Dear Authors,
Thank you for your revisions and for addressing the comments.
Author Response
Dear Authors,
Thank you for your revisions and for addressing the comments.
I appreciate that the authors have justified their stand on the strength of the manuscript. However, I would suggest that the contents are not as per the standards maintained in "Biology" journal. The authors can see for some other journal for it.
RE: Dear reviewer, thank you for appreciating our work!
Reviewer 3 Report
Comments and Suggestions for Authors
I appreciate that the authors have justified their stand on the strength of the manuscript. However, I would suggest that the contents are not as per the standards maintained in "Biology" journal. The authors can see for some other journal for it.
Author Response
I appreciate that the authors have justified their stand on the strength of the manuscript. However, I would suggest that the contents are not as per the standards maintained in "Biology" journal. The authors can see for some other journal for it.
RE: Dear reviewer, we respect your position, however we have made all the changes requested by the reviewers. We have tried to better integrate the text with additional information, without neglecting the limitations of the study related to the small number of cases. However, we hope that the effort we have made to apply information on stage and recurrence on population data in a Tumor Registry that does not routinely collect this information will be appreciated. The added value is precisely that of confirming data from clinical studies with hospital data on population data.